# A Defucosylated Anti-EpCAM Monoclonal Antibody (EpMab-37-mG_2a_-f) Exerts Antitumor Activity in Xenograft Model

**DOI:** 10.3390/antib11040074

**Published:** 2022-11-24

**Authors:** Teizo Asano, Tomohiro Tanaka, Hiroyuki Suzuki, Guanjie Li, Tomokazu Ohishi, Manabu Kawada, Takeo Yoshikawa, Mika K. Kaneko, Yukinari Kato

**Affiliations:** 1Department of Antibody Drug Development, Tohoku University Graduate School of Medicine, 2-1 Seiryo-machi, Aoba-ku, Sendai 980-8575, Japan; 2Department of Molecular Pharmacology, Tohoku University Graduate School of Medicine, 2-1 Seiryo-machi, Aoba-ku, Sendai 980-8575, Japan; 3Institute of Microbial Chemistry (BIKAKEN), Numazu, Microbial Chemistry Research Foundation, 18-24 Miyamoto, Numazu-shi 410-0301, Japan; 4Laboratory of Oncology, Institute of Microbial Chemistry (BIKAKEN), Microbial Chemistry Research Foundation, 3-14-23 Kamiosaki, Shinagawa-ku 141-0021, Tokyo, Japan; 5Department of Pharmacology, Tohoku University Graduate School of Medicine, 2-1 Seiryo-machi, Aoba-ku, Sendai 980-8575, Japan

**Keywords:** EpCAM, breast cancer, pancreatic cancer, antitumor activities, antibody-dependent cellular cytotoxicity

## Abstract

The epithelial cell adhesion molecule (EpCAM) is a stem cell and carcinoma antigen, which mediates cellular adhesion and proliferative signaling by the proteolytic cleavage. In contrast to low expression in normal epithelium, EpCAM is frequently overexpressed in various carcinomas, which correlates with poor prognosis. Therefore, EpCAM has been considered as a promising target for tumor diagnosis and therapy. Using the Cell-Based Immunization and Screening (CBIS) method, we previously established an anti-EpCAM monoclonal antibody (EpMab-37; mouse IgG_1_, kappa). In this study, we investigated the antibody-dependent cellular cytotoxicity (ADCC), complement-dependent cytotoxicity (CDC), and an antitumor activity by a defucosylated mouse IgG_2a_-type of EpMab-37 (EpMab-37-mG_2a_-f) against a breast cancer cell line (BT-474) and a pancreatic cancer cell line (Capan-2), both of which express EpCAM. EpMab-37-mG_2a_-f recognized BT-474 and Capan-2 cells with a moderate binding-affinity [apparent dissociation constant (*K*_D_): 2.9 × 10^−8^ M and 1.8 × 10^−8^ M, respectively] by flow cytometry. EpMab-37-mG_2a_-f exhibited ADCC and CDC for both cells by murine splenocytes and complements, respectively. Furthermore, administration of EpMab-37-mG_2a_-f significantly suppressed the xenograft tumor development compared with the control mouse IgG. These results indicated that EpMab-37-mG_2a_-f exerts antitumor activities and could provide valuable therapeutic regimen for breast and pancreatic cancers.

## 1. Introduction

EpCAM is a unique type I transmembrane glycoprotein which is expressed on the basolateral membrane of epithelial cells [1]. EpCAM mediates homophilic and intercellular adhesion through the extracellular domain, which is essential for the epithelial integrity [2]. EpCAM was the first identified human tumor antigen [3], and the expression is correlated with poor prognosis in various tumors [4,5,6,7]. EpCAM also functions as a signaling molecule. The formation in intercellular EpCAM oligomers triggers the transmembrane proteolytic cleavage by a membrane protease complex. The EpCAM intracellular domain serves as a transcriptional cofactor by interacting with β-catenin, and regulates the transcriptional targets involved in cell proliferation, survival, and stemness [8]. Therefore, the overexpression of EpCAM plays a critical role in the malignant progression of tumors.

Circulating tumor cells (CTCs) are an important indicator of micro-metastasis and provide prognostic and therapeutic information [9]. CellSearch^®^ (Menarini Silicon Biosystems, Inc., Huntington, PA, USA) is a platform that captures EpCAM-positive circulating tumor cells (CTCs) from whole blood samples [10]. The U.S. Food and Drug Administration (FDA) approved CellSearch^®^ in 2009. The EpCAM-based CellSearch^®^ CTC test has been studied in several clinical trials in lung [11], pancreatic [12], and breast [13] cancers.

The clinicopathological classiffications of breast cancer are based on the expression of estrogen receptor (ER), progesteron receptor (PR), human epidermal growth factor receptor 2 (HER2), and Ki-67. The intrinsic subtypes are classified into luminal A (ER+ and/or PR+, HER2−, and low Ki-67), luminal B HER2-negative (ER+ and/or PR+, HER2−, and high Ki-67), luminal B HER2-positive (ER+ and/or PR+, HER2+), HER2-positive non-luminal (ER− and PR−, HER2+), and triple negative (ER−, PR−, HER2−) [14]. EpCAM expression in breast cancer was previously analyzed and showed varied clinical outcomes in the intrinsic subtypes. EpCAM expression was especially associated with an unfavorable prognosis in the luminal B HER2-positive and triple negative subtypes [15].

Pancreatic cancer is one of the most aggressive tumors that is difficult to detect in early stages and has poor treatment options. Approximately 80% of pancreatic cancer patients are in late stage when diagnosed and are not amenable to surgical resection. The 5-year survival is poor, estimated at 2% to 9%. Therapeutic options are largely limited to cytotoxic chemotherapy [16]. Serum carbohydrate antigen 19-9 is the only FDA-approved tumor marker to manage pancreatic cancer status [17]. Therefore, it is necessary to develop groundbreaking early diagnostic markers and new drugs with less toxicity. The antitumor effects on pancreatic tumors of several mAbs have been evaluated in pre-clinical studies and clinical trials [18].

EpCAM was the first target of monoclonal antibody (mAb) therapy in humans [19]. Adecatumumab is a human recombinant mAb [20], and exhibited antitumor activity in colorectal and prostate cancers through antibody-dependent cellular cytotoxicity (ADCC) and complement-dependent cytotoxicity (CDC) [21]. Catumaxomab is a bispecific and trifunctional anti-EpCAM/CD3-antibody. Catumaxomab recognizes EpCAM on tumors, recruits T cells through the anti-CD3 arm, and recruits natural killer (NK) cells and macrophages through the Fc domain. These events enhanced ADCC activity [22]. Catumaxomab exhibited promising outcomes in several clinical trials [23,24,25], and was approved by the European Union for the treatment of patients with malignant ascites in 2009 [23].

The composition of the *N*-linked glycans on IgG plays a critical role in effector functions [26]. In other antibody classes, the influence of glycans, particularly sialic acid, have been suggested [27,28]. The binding of the FcγRIIIa on NK cells to the Fc region of mAbs mediates ADCC [29]. However, the *N*-linked glycosylation in the Fc region is reported to impair the binding to the FcγRIIIa on NK cells [30,31]. Fucosyltransferase 8 (FUT8) plays a critical role in the *N*-linked glycosylation (core fucosylation) of the Fc region [31]. Therefore, FUT8-knockout Chinese hamster ovary (CHO) cells have been shown to produce completely defucosylated recombinant antibodies in the Fc region [32]. The defucosylated mAb potently binds to FcγRIIIa and exerts increased ADCC activity compared with a conventional mAb [30].

We previously established an anti-EpCAM mAb, EpMab-37 (mouse IgG_1_, kappa) using the Cell-Based Immunization and Screening (CBIS) method [33]. Because mouse IgG_2a_ can bind to FcγRIV, the murine orthologue of human FcγRIIIa, we produced a subclass-switched and a defucosylated EpMab-37 (EpMab-37-mG_2a_-f) using FUT8-deficient CHO cells. In this study, we investigated the ability of EpMab-37-mG_2a_-f to induce ADCC, CDC, and antitumor activities against a breast cancer cell line (BT-474) derived from the luminal B HER2-positive subtype [34], and a pancreatic cancer cell line (Capan-2), both of which express EpCAM.

## 2. Materials and Methods

### 2.1. Cell Lines

A human breast cancer cell line (BT-474) and a pancreatic cancer cell line (Capan-2) were purchased from the American Type Culture Collection (ATCC, Manassas, VA, USA). BT-474 was cultured in Dulbecco’s Modified Eagle Medium (Nacalai Tesque, Inc. (Nacalai), Kyoto, Japan), supplemented with 10% fetal bovine serum (FBS; Thermo Fisher Scientific, Inc. (Thermo), Waltham, MA, USA), 100 units/mL of penicillin, 100 μg/mL streptomycin, and 0.25 μg/mL amphotericin B (Nacalai). Capan-2 was cultured in McCOY’s medium (Cytiva, Tokyo, Japan), supplemented with 10% FBS, 100 μg/mL streptomycin, 100 units/mL of penicillin, and 0.25 μg/mL amphotericin B. The cell lines were maintained in a humidified atmosphere under 5% CO_2_ at 37 °C.

### 2.2. Animals

All animal experiments were performed following regulations and guidelines to minimize animal distress and suffering in the laboratory by the Institutional Committee for Experiments of the Institute of Microbial Chemistry (Numazu, Japan) (approval no. 2022-024). Mice were maintained on an 11 h light/13 h dark cycle in a specific pathogen-free environment across the experimental period. Food and water were supplied ad libitum. Mice weight was monitored twice per week and health was monitored three times per week. The loss of original body weight was determined to a point >25% and/or a maximum tumor size >2000 mm^3^ and/or significant changes in the appearance of tumors as humane endpoints for euthanasia. Cervical dislocation was used for euthanasia. Mice death was confirmed by respiratory arrest and rigor mortis.

### 2.3. Antibodies

An anti-EpCAM mAb, EpMab-37, was established as previously described [27]. To switch the subclass of EpMab-37 from mouse IgG_1_ to mouse IgG_2a_ (EpMab-37-mG_2a_), we cloned V_H_ cDNA of EpMab-37 and C_H_ of mouse IgG_2a_ into the pCAG-Ble vector (FUJIFILM Wako Pure Chemical Corporation (Wako), Osaka, Japan). V_L_ cDNA of EpMab-37 and C_L_ cDNA of mouse kappa light chain were also cloned into the pCAG-Neo vector (Wako). To generate the defucosylated EpMab-37-mG_2a_, the vector for the EpMab-37-mG_2a_ was transfected into FUT8 knockout ExpiCHO-S (BINDS-09) cells using the ExpiCHO Expression System (Thermo) [35,36,37,38,39,40,41,42,43,44,45,46,47,48,49]. Defucosytaled EpMab-37-mG_2a_ (EpMab-37-mG_2a_-f) was purified using Ab-Capcher (ProteNova Co., Ltd., Kanagawa, Japan). Mouse IgG (cat. no. 140-09511) and IgG_2a_ (cat. no. M7769) were purchased from Wako and Sigma-Aldrich (St. Louis, MO, USA), respectively. A 281-mG_2a_-f (a defucosylated anti-hamster podoplanin [PDPN] mAb, control mouse IgG_2a_ for ADCC reporter assay) was previously described [50]. Trastuzumab was purchased from the R&D systems (Minneapolis, MN, USA).

### 2.4. Flow Cytometry

Cells were obtained by 0.25% trypsin/1 mM ethylenediamine tetraacetic acid (EDTA; Nacalai). Cells were treated with 10 µg/mL of EpMab-37-mG_2a_-f or blocking buffer [0.1% bovine serum albumin (BSA; Nacalai) in phosphate-buffered saline (PBS)] (control) for 30 min at 4 °C. Then, cells were incubated in Alexa Fluor 488-conjugated anti-mouse IgG (1:2000; Cell Signaling Technology, Inc., Danvers, MA, USA) for 30 min at 4 °C. Fluorescence data were collected by the Cell Analyzer EC800, and analyzed by EC800 software ver. 1.3.6 (Sony Corp., Tokyo, Japan).

### 2.5. Determination of Binding Affinity

Serially diluted EpMab-37-mG_2a_-f (0.006–100 μg/mL) was suspended with BT-474 and Capan-2 cells. Then, the cells were further treated with Alexa Fluor 488-conjugated anti-mouse IgG (1:200). Fluorescence data were obtained by BD FACSLyric (BD Biosciences, San Jose, CA, USA). To determine the dissociation constant (*K*_D_), the fitting binding isotherms to built-in one-site binding models in GraphPad Prism 9 (GraphPad Software, La Jolla, CA, USA) was used.

### 2.6. ADCC

ADCC induction by EpMab-37-mG_2a_-f was assayed as follows. Six female BALB/c nude mice (five-week-old) were purchased from Charles River Laboratories, Inc. Spleens were aseptically removed, and single-cell suspensions were obtained through a sterile cell strainer (352360, BD Falcon). Erythrocytes were removed with treatment of ice-cold distilled water. The splenocytes were resuspended in the medium; this preparation was designated as effector cells. Target cells (BT-474 and Capan-2) were treated with Calcein AM (10 μg/mL, Thermo Fisher Scientific, Inc.). The target cells (2 × 10^4^ cells) were mixed with effector cells (effector-to-target ratio, 100:1), 100 μg/mL of EpMab-37-mG_2a_-f or control mouse IgG_2a_ in 96-well plates. After incubation for 4.5 h at 37 °C, the Calcein release into the medium was measured with an excitation wavelength (485 nm) and an emission wavelength (538 nm) using a microplate reader (Power Scan HT; BioTek Instruments, Inc., Winooski, VT, USA).

Cytolyticity (% lysis) was determined as follows: % lysis = (E − S)/(M − S) × 100. “E” is the fluorescence in the presence of both effector and target cells. “S” is the spontaneous fluorescence in the presence of only target cells. “M” is the maximum fluorescence by the treatment with a lysis buffer (10 mM Tris-HCl (pH 7.4), 10 mM of EDTA, and 0.5% Triton X-100).

### 2.7. CDC

BT-474 and Capan-2 cells were treated with Calcein-AM (10 μg/mL), resuspended in medium and plated in 96-well plates, at 2 × 10^4^ cells/well, with 10% rabbit complement (Low-Tox-M rabbit complement; Cedarlane Laboratories, Hornby, ON, Canada), 100 μg/mL of EpMab-37-mG_2a_-f, or control mouse IgG_2a_ added to each well. Following incubation for 4.5 h at 37 °C, Calcein release into the medium was determined.

### 2.8. ADCC Reporter Bioassay

The ADCC reporter bioassay was performed using an ADCC Reporter Bioassay kit from Promega (Madison, WI, USA), following the manufacturer’s instructions [51]. Target cells (12,500 cells per well) were inoculated into a 96 well white solid plate. EpMab-37-mG_2a_-f, EpMab-37, and 281-mG_2a_-f were serially diluted and added to the target cells. Jurkat cells stably expressing the human FcγRIIIa receptor, and a Nuclear Factor of Activated T-Cells (NFAT) response element driving firefly luciferase, were used as effector cells. The engineered Jurkat cells (75,000 cells in 25 μL) were then added and co-cultured with antibody-treated target cells at 37 °C for 6 h. Luminescence using the Bio-Glo Luciferase Assay System (Promega) was measured with a GloMax luminometer (Promega).

### 2.9. Antitumor Activities in Xenografts of BT-474 and Capan-2

BALB/c nude mice (female, 4 weeks old) were purchased from Charles River Laboratories, Inc. The cells (5 × 10^6^ cells) were resuspended with BD Matrigel Matrix Growth Factor Reduced (BD Biosciences) and were subcutaneously injected into the left flank of 10 weeks old mice. On day 6 post-inoculation, 100 μg of EpMab-37-mG_2a_-f (*n* = 8) or control mouse IgG (*n* = 8) in 100 μL PBS were injected intraperitoneally. On day 13 and 18, additional antibodies were injected. The tumor volume was measured on days 6, 11, 13, 18, 22, and 24 after the injection of cells. Tumor volumes were determined as previously described [35,36,37,38,39,40,41,42,43,44,45,46,47,48,49,52,53,54,55].

### 2.10. Statistical Analysis

All data are represented as mean ± standard error of the mean (SEM). Welch’s *t* test was conducted for ADCC activity, CDC activity, and tumor weight. ANOVA with Sidak’s post hoc test was conducted for tumor volume and mouse weight. GraphPad Prism 8 (GraphPad Software, Inc.) was used for all calculations. *p* < 0.05 was considered to indicate a statistically significant difference.

## 3. Results

### 3.1. Flow Cytometric Analysis

In our previous study, a sensitive and specific anti-EpCAM mAb, EpMab-37 (mouse IgG_1_, kappa), was established using the CBIS method [33]. We first performed flow cytometric analysis using EpMab-37-mG_2a_-f against BT-474 and Capan-2 cells and found that EpMab-37-mG_2a_-f recognized both cells (Figure 1A). A kinetic analysis of the interactions of EpMab-37-mG_2a_-f with BT-474 and Capan-2 cells was conducted using flow cytometry. The apparent dissociation constants (*K*_D_) of EpMab-37-mG_2a_-f to BT-474 and Capan-2 cells were 2.9 × 10^−8^ M and 1.8 × 10^−8^ M, respectively (Figure 1B), indicating moderate binding affinity of EpMab-37-mG_2a_-f against both cells.

### 3.2. ADCC and CDC Activities of EpMab-37-mG_2a_-f in BT-474 and Capan-2

We next examined whether EpMab-37-mG_2a_-f induces ADCC and CDC activities in BT-474 and Capan-2 cells. EpMab-37-mG_2a_-f exhibited much higher ADCC (64.0% cytotoxicity) in BT-474 cells than that of control mouse IgG_2a_ (8.49% cytotoxicity; *p* < 0.01) (Figure 2A). EpMab-37-mG_2a_-f also showed higher CDC activity (64.2% cytotoxicity) in BT-474 cells than that of control mouse IgG_2a_ (11.1% cytotoxicity; *p* < 0.01) (Figure 2B). Furthermore, EpMab-37-mG_2a_-f showed that ADCC (24.8% cytotoxicity) against Capan-2 cells were more potent than control mouse IgG_2a_ (5.6% cytotoxicity; *p* < 0.01) (Figure 2C). EpMab-37-mG_2a_-f elicited a higher degree of CDC (29.3% cytotoxicity) in Capan-2 cells compared with that elicited by control mouse IgG_2a_ (6.8% cytotoxicity; *p* < 0.05) (Figure 2D). These results demonstrated that EpMab-37-mG_2a_-f exhibited the ADCC and CDC activities against BT-474 and Capan-2 cells.

The ADCC reporter bioassay is a bioluminescent reporter gene assay to quantify the biological activity of the antibody via FcγRIIIa-mediated pathway activation in an ADCC mechanism of action [51]. To compare the ADCC pathway activation by EpMab-37-mG_2a_-f, we treated BT-474 and Capan-2 cells with serially diluted mAbs, and then incubated with effector Jurkat cells, which express the human FcγRIIIa receptor and an NFAT response element driving firefly luciferase. Furthermore, we also used defucosylated anti-hamster PDPN mAb (mouse IgG_2a_) (281-mG_2a_-f) as a control since we could not obtain defucosylated control normal mouse IgG_2a_. We confirmed that 281-mG_2a_-f never recognized the target cells. As shown in Figure 3A, EpMab-37-mG_2a_-f dose-dependently activated the effector (EC_50_; 45.8 ng/mL), but 281-mG_2a_-f did not. Similar results were obtained in the presence of Capan-2 cells (EC_50_; 10.9 ng/mL) (Figure 3B). We confirmed that original EpMab-37 never activated the effector cells (Appendix A). Furthermore, we compared the ADCC effector activation by EpMab-37-mG_2a_-f with an existing therapeutic antibody, Trastuzumab. We used the target, BT-474 cells which is known as HER2 positive breast cancer. EpMab-37-mG_2a_-f exhibited a superior EC_50_ compared to trastuzumab (Appendix A). These results indicated that EpMab-37-mG_2a_-f exhibits superior ADCC activation activity compared with EpMab-37 and trastuzumab.

### 3.3. Antitumor Effects of EpMab-37-mG_2a_-f in Mouse Xenograft Models of BT-474 and Capan-2

BT-474 cells were inoculated into the left flank of mice, followed by the intraperitoneally injection of EpMab-37-mG_2a_-f or control mouse IgG on days 6, 13, and 18. The tumor volume was measured after the inoculation. EpMab-37-mG_2a_-f-treated mice exhibited significantly less tumor volume on day 11 (*p* < 0.01), day 13 (*p* < 0.01), day 18 (*p* < 0.01), day 22 (*p* < 0.01), and day 24 (*p* < 0.01), compared with control mouse IgG-treated control mice (Figure 4A). The EpMab-37-mG_2a_-f treatment resulted in a 51.3% reduction of the tumor volume compared with that of the control mouse IgG on day 24 (Figure 4C). Tumors from EpMab-37-mG_2a_-f-treated mice weighed significantly less than tumors from control IgG-treated control mice (52.4% reduction, *p* < 0.01; Figure 4E). Resected tumors on day 24 are presented in Figure 4E. The total body weights did not significantly differ between the EpMab-37-mG_2a_-f-treatment and the control groups (Figure 5A). The body appearance of mice on day 24 post inoculation is shown in Figure 5C, and the body weights’ loss and skin disorder were not observed.

In the Capan-2 xenograft, EpMab-37-mG_2a_-f and control mouse IgG were injected intraperitoneally into mice on days 6, 14, and 18 after the inoculation of Capan-2 cells. The tumor volume was measured on days 6, 11, 14, 18, 20, 25, and 27. The treatment of EpMab-37-mG_2a_-f resulted in a significant inhibition in tumor development on days 11 (*p* < 0.05), 14 (*p* < 0.01), 18 (*p* < 0.01), 20 (*p* < 0.01), 25 (*p* < 0.01), and 27 (*p* < 0.01) compared with that of the control mouse IgG (Figure 4B). The administration of EpMab-37-mG_2a_-f resulted in a 51.3% reduction of tumor volume compared with that of the control mouse IgG on day 27. Furthermore, the tumor weight of the EpMab-37-mG_2a_-f -treated mice was significantly decreased than that of control IgG-treated mice (49.1% reduction; *p* < 0.01, Figure 4D). The excised tumors of control and EpMab-37-mG_2a_-f -treated groups on day 27 are shown in Figure 4F. Total body weights were almost similar in both groups (Figure 5B). Appearance of mice on day 27 after inoculation of cells are shown in Figure 5D.

These results indicate that administration of EpMab-37-mG_2a_-f effectively suppresses the tumor growth of BT-474 and Capan-2 xenografts.

## 4. Discussion

The impact of EpCAM expression on breast cancer prognosis is dependent on intrinsic subtype. In the luminal B HER2-positive and triple negative subtypes, EpCAM expression is associated with an unfavorable prognosis. In contrast, EpCAM expression is associated with a favorable prognosis in the HER2-positive non-luminal subtype [15]. Therefore, the luminal B HER2-positive and triple negative subtypes are potential groups for treatment with EpCAM-targeting therapy. In this study, we investigated the antitumor effect of a defucosylated anti-EpCAM mAb (EpMab-37-mG_2a_-f) against a breast cancer cell line, BT-474 derived from luminal B HER2-positive subtype [34]. EpMab-37-mG_2a_-f exhibited superior ADCC and CDC activities in vitro (Figure 2 and Figure 3), and antitumor activity against BT-474 xenograft in nude mice (Figure 4). We previously developed an anti-HER2 mAb (H_2_Mab-19) and examined ADCC, CDC, and antitumor activities against BT-474 cells [45]. Although the binding affinity of H_2_Mab-19 and EpMab-37-mG_2a_-f to BT-474 cells were comparable, EpMab-37-mG_2a_-f exerted more potent ADCC activity and antitumor effect in vivo. These results are probably due to the defucosylation in EpMab-37-mG_2a_-f, but not in H_2_Mab-19. Moreover, EpCAM forms a cis-dimer which further makes a biologically relevant oligomeric state (e.g., cis-tetramer, trans-tetramer, and trans-octamer) according to several experimental observations [56]. These oligomeric structures of EpCAM could promote the clustering of anti-EpCAM mAbs, which might help the FcγRIIIa engagement on effector cells, and potentiate the ADCC activity.

EpCAM is an important cell surface molecule to collect CTCs [10]. In a prospective study of pancreatic cancer patients, the CTC in the peripheral blood affect the outcome of patients independent from other risk factors, including adjuvant chemotherapy [12]. Furthermore, EpCAM-sorted pancreatic adenocarcinoma cells from surgically resected tumors could be applied to the analysis of tumor-cell-intrinsic chromatin accessibility patterns. A chromatin accessibility signature and associated transcriptional factors (ZKSCAN1 and HNF1β) are significantly correlated with pancreatic cancer prognosis [57]. This information could contribute to the selection of patients to be applied to anti-EpCAM mAb therapy. Recently, CTC expansion techniques have been developed to evaluate the characteristics of CTCs. The techniques include the two-dimensional (2D) long-term expansion, 3D organoids/spheroids culture, and in vivo xenografts/metastasis formation in immunodeficient mice [58]. It would be worthwhile to investigate the effect of EpMab-37-mG_2a_-f on the 2D and 3D CTC expansion in vitro and anti-metastatic activity in vivo.

We have been investigating the critical epitope of EpMab-37 [33], and recently identified that Arg163 of EpCAM is the most important residue of the EpMab-37 epitope (submitted). Among clinically tested mAbs, no mAb recognized above region, suggesting that EpMab-37 possesses a unique epitope and a different mode of actions. In this study, we did not examine the EpCAM-internalizing activity by EpMab-37-mG_2a_-f. Furthermore, the relationship between the internalizing activity and the epitope has not been investigated. In future study, we would like to evaluate it for the development of antibody-drug conjugates.

Anti-EpCAM mAb can be used for a bispecific Ab with anti-MET mAb [59]. MM-131 is a bispecific Ab that is monovalent against MET, but it exhibits high avidity to EpCAM through binding to a single chain Fv of an anti-EpCAM mAb, MOC31 [60]. MM-131 exhibits antagonistic activity that interferes both ligand-dependent and ligand-independent MET signaling and induces the receptor down-regulation [59]. MCLA-128 is a bispecific Ab for HER2 and HER3. MCLA-128 can inhibit heregulin (a HER3 ligand)-mediated signaling of HER2/HER3 heterodimer and suppress tumor cell growth via the suppression of PI3K/Akt signaling [61]. Clinical studies on MCLA-128 are ongoing in patients with breast cancer, pancreatic cancer, and non-small cell lung cancers [62]. In the future study, we would like to apply EpMab-37 for the combination therapy with anti-HER2 mAbs or develop a bispecific mAb targeting EpCAM and HER2.

## Figures and Tables

**Figure 1 antibodies-11-00074-f001:**
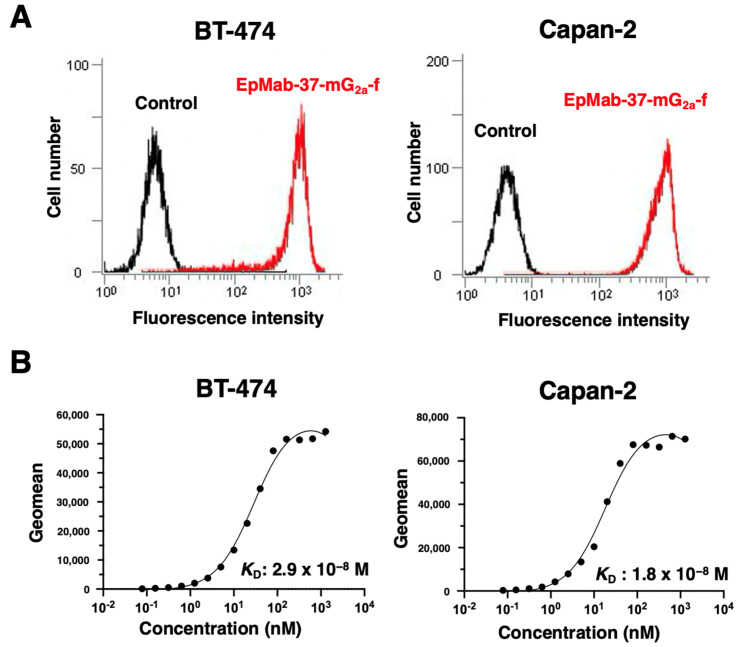
Flow cytometric analyses. (**A**) BT-474 and Capan-2 cells were treated with 10 µg/mL of EpMab-37-mG_2a_-f (red line) or blocking buffer as a negative control (black line), followed by addition of Alexa Fluor 488-conjugated anti-mouse IgG. Data were collected using the EC800 Cell Analyzer. (**B**) BT-474 and Capan-2 cells were suspended in 100 μL of serially diluted EpMab-37-mG_2a_-f (0.006–100 μg/mL), followed by the addition of Alexa Fluor 488-conjugated anti-mouse IgG. Data were collected using the BD FACS Lyric and analyzed by GraphPad PRISM 9 to determine the apparent dissociation constant (*K*_D_).

**Figure 2 antibodies-11-00074-f002:**
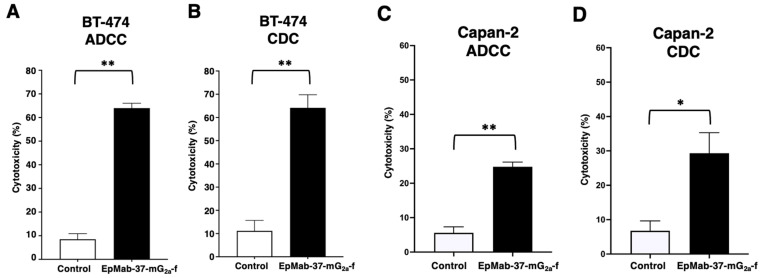
Evaluation of EpMab-37-mG_2a_-f mediated ADCC and CDC activities in BT-474 and Capan-2 cells. (**A**,**C**) ADCC elicited by EpMab-37-mG_2a_-f or control mouse IgG_2a_ against BT-474 (**A**) and Capan-2 (**C**) cells. (**B**,**D**) CDC elicited by EpMab-37-mG_2a_-f or control mouse IgG_2a_ against BT-474 (**B**) and Capan-2 (**D**) cells. Values are shown as mean ± SEM. Asterisks indicate statistical significance (** *p* < 0.01, * *p* < 0.05; Welch’s *t*-test). ADCC, antibody-dependent cellular cytotoxicity; CDC, complement-dependent cytotoxicity.

**Figure 3 antibodies-11-00074-f003:**
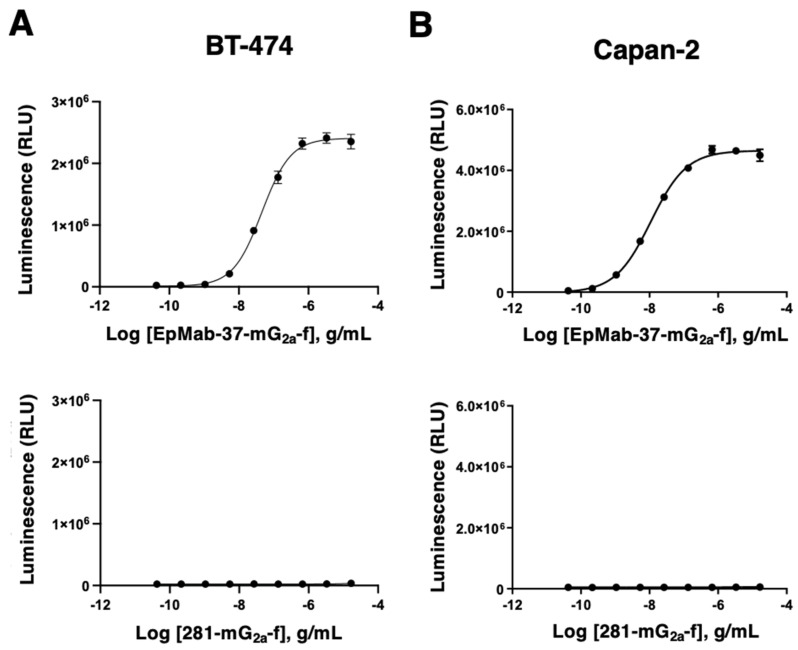
ADCC reporter assay by EpMab-37-mG_2a_-f and 281-mG_2a_-f in the presence of BT-474 and Capan-2 cells. (**A**,**B**) Target cells (BT-474 (**A**) and Capan-2 (**B**)) were treated with serially diluted EpMab-37-mG_2a_-f and 281-mG_2a_-f (mouse IgG_2a_ control). Then, Jurkat cells stably expressing the human FcγRIIIa receptor and an NFAT response element driving firefly luciferase were added and co-cultured. Luminescence was measured using the Bio-Glo Luciferase Assay System. ADCC, antibody-dependent cellular cytotoxicity.

**Figure 4 antibodies-11-00074-f004:**
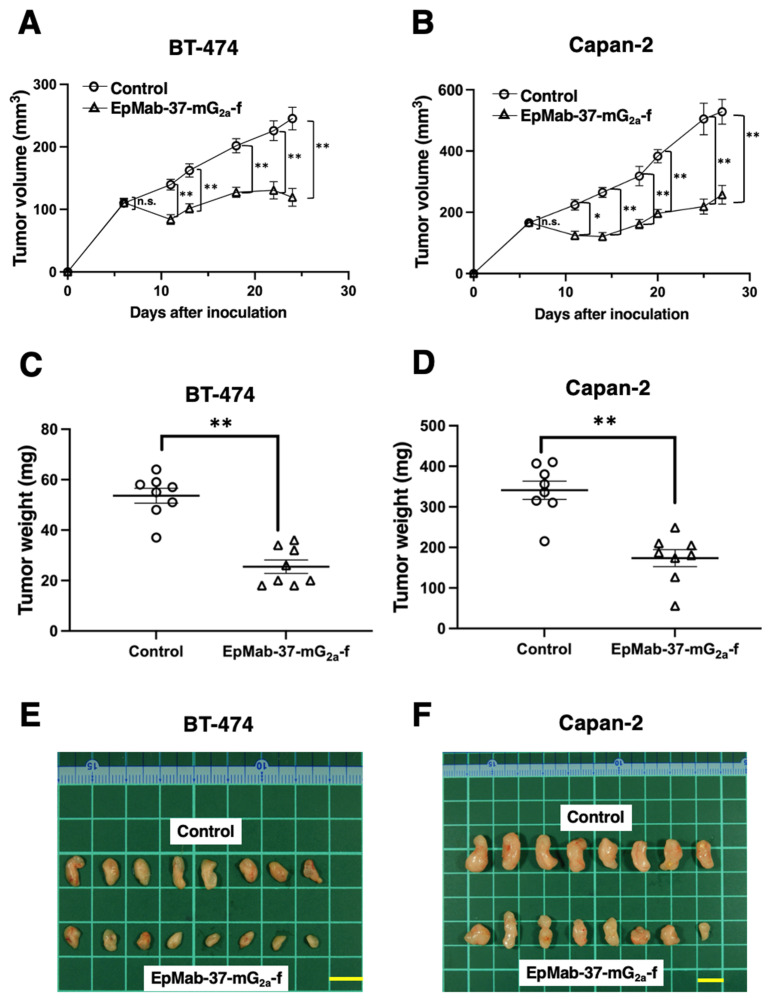
Antitumor activity of EpMab-37-mG_2a_-f. (**A**,**B**) Measurement of tumor volume in (**A**) BT-474 and (**B**) Capan-2 xenograft models. BT-474 and Capan-2 cells (5 × 10^6^ cells) were inoculated into mice subcutaneously. On day six, 100 μg of EpMab-37-mG_2a_-f or control mouse IgG were injected into mice intraperitoneally. On day 13 and 18 (BT-474) or 14 and 18 (Capan-2), additional antibodies were injected. On the indicated days after the inoculation, the tumor volume was measured. Values are presented as the mean ± SEM. ** *p* < 0.01, * *p* < 0.05 (ANOVA and Sidak’s multiple comparisons test). (**C**,**D**) The weight of excised (**C**) BT-474 and (**D**) Capan-2 xenografts was measured on day 24 and 27, respectively. Values are presented as the mean ± SEM. ** *p* < 0.01 (Welch’s *t* test). (**E**,**F**) The resected tumors appearance of (**E**) BT-474 and (**F**) Capan-2 xenografts in the control mouse IgG and EpMab-37-mG_2a_-f treated groups on day 24 and 27, respectively (scale bar, 1 cm). n.s., not significant.

**Figure 5 antibodies-11-00074-f005:**
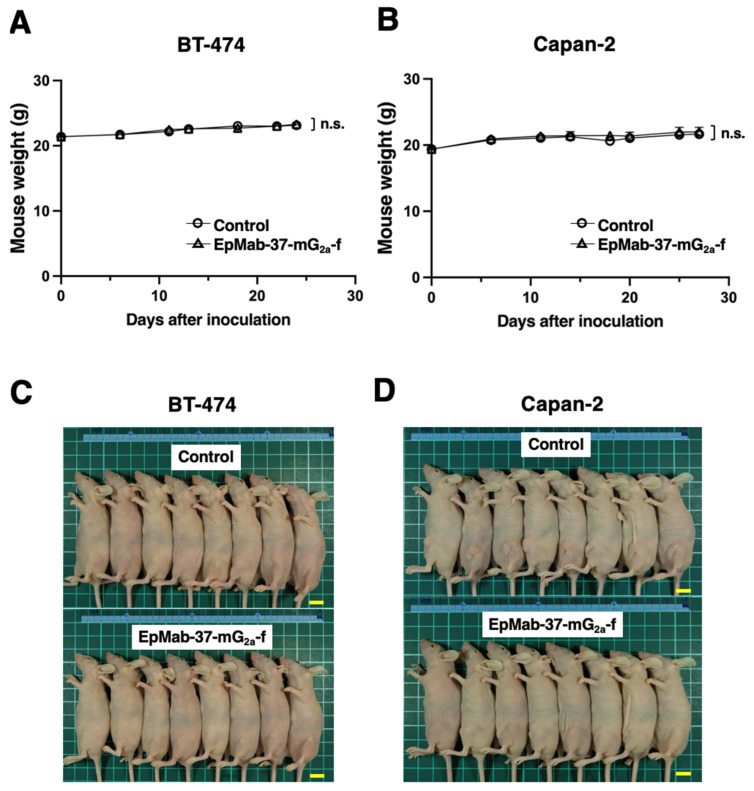
Mice body weights and appearance. (**A**,**B**) Body weights in (**A**) BT-474 and (**B**) Capan-2 xenografts-implanted mice on the indicated days (ANOVA and Sidak’s multiple comparisons test). (**C**,**D**) Body appearance in (**C**) BT-474 and (**D**) Capan-2 xenografts-implanted mice on day 24 and 27, respectively (scale bar, 1 cm). n.s., not significant.

## Data Availability

Data is contained within the article or Appendix A.

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
