# Peer review of "A Defucosylated Anti-EpCAM Monoclonal Antibody (EpMab-37-mG2a-f) Exerts Antitumor Activity in Xenograft Model"

_2073-4468, 2022, doi:10.3390/antib11040074_

Round 1

Reviewer 1 Report

The paper's focus is on antibody afucosylation increases ADCC and CDC which is well established across human and mice species. 

1)    Requires significant changes to the introduction. Work is focused on antibody glycosylation, but the glycosylation role is not mentioned.  Authors should include antibody glycosylation's role in antibody structure and function. Each sugar plays an important role in mediating function. Here is a recent review on antibody glycosylation, 

https://www.ncbi.nlm.nih.gov/pmc/articles/PMC9021442/, another review exclusively focused on a fucosylation https://pubmed.ncbi.nlm.nih.gov/29733746/. Discuss and cite these papers.

2)    Instead of defucosylation, it is Afucosylation as mentioned in other papers. https://www.ncbi.nlm.nih.gov/pmc/articles/PMC6150623/, https://pubmed.ncbi.nlm.nih.gov/22305040/. Show different types of mouse glycans and their functions similar to https://www.ncbi.nlm.nih.gov/pmc/articles/PMC9021442/

3)    Afucosyltaion should be included in the title because increased ADCC and CDC are resulting from Afucosylation

4)    Authors did not show a result antibody is afucosylated. It is a common practice to show HPLC or mass spec results to show the absence of fucose.

5)    One of the major drawbacks in experiment design is using of mouse IgG2a control. In the previous paper, authors showed EpMab-37-mG2a anti-tumor activity. The author should have used EpMab-37-mG2a to show the difference in increased ADCC as a result of the removal of fucose. 

6)    Authors did not discuss Fc receptors and the role of fucose in increasing ADCC and CDC.

7)    Show a diagram with afucosylated antibody triggering ADCC through Fc receptors.

Reviewer 2 Report

Minor comments:

1.       What are the merits of this monoclonal antibody?  The author has mentioned it in line 251-252, please extend this part in discussion and provide readers with more information why you develop this antibody.

2.       Why did you do the sub-class switch from mG1 to mG2a. Please provide readers with more information either in introduction or discussion.

3.       Why use nude mice as the ADCC model, please give us more background information. Is it NK or macrophage or any others cells that paly a role in this model?

Major comments:

Only one cell line was used in this study, which I don’t think is enough. Please repeat figure 2 in other cell lines and see if the results are reproducible.

Reviewer 3 Report

Asano Etal, present a research article describing some of the biological activity of an Anti-EpCAM antibody termed EpMab-37-mG2a-f.

The authors present data from flow cytometry staining, ADCC, CDC and murine Xenograft model which show activity when compared against an isotype IgG2a control antibody. In the Xenograft model, the antibody inhibited tumor growth as evidenced by slower growth of the BT474 tumor xenograft when compared to the control antibody.

These data suggest that the reformatted antibody in IgG2a format retains its biological activity. However, the purpose of re-formatting or the use of such re-formatting was not mentioned. Since it was previously established that the IgG1 version of the antibody is biologically active, an apt set of experiments would have been a direct comparison of the IgG2a version of the antibody to that of the original format or to another IgG2a antibody that targets the EpCAM protein target. It would have been more impactful to compare the EpMab-37-mG2a-f to the currently available therapeutic antibody(ies).

No further work to characterize the antibody was performed. Experiments such as epitope mapping could be very informative if this antibody is different from the currently available clinical anti-EPCAM antibodies or if it holds any advantage over the clinical antibody.

To determine the affinity of the antibody, the authors used a flow cytometry-based assay and determined the kd to be 29 nM. Here this approach is limited in determining the actual affinity as different cell lines may express EpCAM at different levels. The antibody is not directly conjugated to the fluorescent tag and a secondary antibody was used, where signal amplification could occur which adds a layer of complexity. A better approach for using flow cytometry would have been to use an scFv directly conjugated to a fluorescent label. This would have resulted in a 1:1 binding and therefore may reflect true affinity. The other more commonly used approaches such as SPR and Octet could be used to compare the kinetic profile of the antibody as well.

In the ADCC and CDC assays, a very high ratio of effector to target cells (100 : 1) and a very high antibody concentration were used (100 µg/ml). The authors may have chosen this high ratio and concentration to account for the weak affinity of the antibody. Since no comparison to the original IgG1 version or a therapeutic anti-EPCAM antibody was not made, the ADC and CDC data offer little insight into the antibody efficacy or its advantages over existing therapeutic antibodies.

Overall, the current manuscript is easy to read and understand, the data is presented in a visually easy-to-understand format. However, the current publication lacks novelty and does not present any evidence about the significance of the antibody being developed or its advantages and falls short in both adressing the gaps in knowledge, therapy, mode of action, etc of EpCAM or any therapeutic antibodies targeting EpCam. In the discussion, the authors talk about critical epitope and propose that the Arg163 residue is essential EpMab-37-mG2a-f (finally a difference from other anti-EpCAM antibodies). The data which shows the role Arg163 plays in EpCAM’s pro-oncogenic nature would have made a more compelling manuscript. I look forward to reading that manuscript. Finally, for the current manuscript should include a direct comparison to the original IgG1 version, as this is an important control to show biological equivalance or to high light any differences. The reasons for the re-formatting etc should also be mentioned.

Round 2

Reviewer 1 Report

1) In the line 84 to 85 mentioned “However, the N-linked glycosylation in the Fc region is reported to impair the binding to the FcγRIIIa on NK cells”. Is it impair or increases?

2) Thank you for adding Antibody glycosylation in the introduction but does not add details such as which residue undergoes glycosylation and what about sialic acid. I understand the manuscript is focused on fucose, but We should also keep audience in mind to show that each sugar plays an important role in regulating antibody function. Discuss following papers in in 2 or 3 line, https://www.science.org/doi/full/10.1126/science.1129594 https://www.frontiersin.org/articles/10.3389/fimmu.2022.818736/full

3) Authors used 281-mG2a-f as control. What is the variable region of the antibody? could we compare the antibody with two different variable regions (281-mG2a-f  Vs EpMab-37-mG2a-f) . Does this control binds to the cells? If it is not binding how, it would trigger ADCC? May be using mG2 a aginist BT4 could be a better control?

4) In figure 1 authors shown EpMab-37-mG2a-f binds to the BT-474 and Capan-2 cells. How about showing the 281-mG2a-f binding affinity. If the control antibody does not engage the target cells it will not trigger the ADCC.

5) Authors switched between two controls 281-mG2a-f and control mG2 a against BT4 for showing control removal of fucose?.

6) In supplemental figure 1, what is the subclass of EpMab-37?. Why this control not tested with the BT-474 cells ? Does EpMab-37 binds to the capan 2 cells ?

7) Comparing EpMab-37mG2a-F with EpMab-37mG2a is better approach to show the role of fucose. Check the following paper shown role of each glycan in Rituximab ADCC https://www.pnas.org/doi/10.1073/pnas.1702173114

8) Could authors show binding of controls to the cells like figure 1.

Reviewer 2 Report

I am satisfied with the response of the authors. 

Author Response

Thank you for the acceptance of revision.

Reviewer 3 Report

Asano Et al, in the revised version, attempted to address the reviewer's concerns about comparing the re-formatted antibody " defucosylated mouse IgG2a-type of 27 EpMab-37 (EpMab-37-mG2a-f)" with the original IgG1 EPMab-37, an anti-HER2 antibody therapeutic Trastuzumab, and 281-mG2a-f defucosylated control antibody.  These additional controls support the notion that EpMab-37-mG2a-f shows stronger activity over the control antibodies in the limited cell lines tested.  The limited variety of cancer cell lines used here are still not representative of how complex the pancreatic and breast cancers could be with regards to EpCAM expression, for example MiaPaCa-2 cells show little expression of EpCam vs L3.6pL cell line which shows much higher levels of protein expression (Meng et al PMID: 26885196) , however, the extensive screening against several different cancer cell types or cell lines may be necessary only if the current version of the antibody is being proposed as a therapeutic option for people.

The author's responses to the question on affinity studies via flow cytometry are acceptable. But other orthogonal methods such as SPR, BLI, MST or Ligand tracer based affinity characterization should be carried out.

Could the authors provide any data that supports the claim that the current version of the antibody is defucosylated?
